# New Frontiers for the Use of IP6 and Inositol Combination in Treating Diabetes Mellitus: A Review

**DOI:** 10.3390/molecules25071720

**Published:** 2020-04-10

**Authors:** Felix O. Omoruyi, Dewayne Stennett, Shadae Foster, Lowell Dilworth

**Affiliations:** 1Department of Life Sciences, Texas A&M University, Corpus Christi, TX 78412, USA; felix.omoruyi@tamucc.edu; 2Department of Basic Medical Sciences, The University of the West Indies Mona Campus, Kingston 7, Mona, Jamaica; Dstenno@yahoo.com (D.S.); shadae.foster@yahoo.com (S.F.); 3Department of Pathology, The University of the West Indies Mona Campus, Kingston 7, Mona, Jamaica

**Keywords:** Ip6, inositol, diabetes mellitus, myo-inositol, lipid, free radical

## Abstract

Inositol, or myo-inositol, and associated analog molecules, including myo-inositol hexakisphosphate, are known to possess beneficial biomedical properties and are now being widely studied. The impact of these compounds in improving diabetic indices is significant, especially in light of the high cost of treating diabetes mellitus and associated disorders globally. It is theorized that, within ten years, the global population of people with the disease will reach 578 million individuals, with the cost of care projected to be approximately 2.5 trillion dollars. Natural alternatives to pharmaceuticals are being sought, and this has led to studies involving inositol, and myo-inositol-hexakisphosphate, also referred to as IP6. It has been reported that IP6 can improve diabetic indices and regulate the activities of some metabolic enzymes involved in lipid and carbohydrate metabolism. Current research activities have been focusing on the mechanisms of action of inositol and IP6 in the amelioration of the indices of diabetes mellitus. We demonstrated that an IP6 and inositol combination supplement may regulate insulin secretion, modulate serum leptin concentrations, food intake, and associated weight gain, which may be beneficial in both prediabetic and diabetic states. The supplement attenuates vascular damage by reducing red cell distribution width. Serum HDL is increased while serum triglycerides tend to decrease with consumption of the combination supplement, perhaps due to the modulation of lipogenesis involving reduced serum lipase activity. We also noted increased fecal lipid output following combination supplement consumption. Importantly, liver function was found to be preserved. Concurrently, serum reactive oxygen species production was reduced, indicating that inositol and IP6 supplement consumption may reduce free radical damage to tissues and organs as well as serum lipids and blood glucose by preserving liver function. This review provides an overview of the findings associated with inositol and IP6 supplementation in the effective treatment of diabetes with a view to proposing the potential mechanisms of action.

## 1. Introduction

Diabetes mellitus is a metabolic disease that is characterized by hyperglycemia with associated alterations in carbohydrate, protein, and fat metabolism. It is an endocrine disorder that presents severe clinical and public health problems globally. Currently, over 463 million individuals are living with diabetes, and its prevalence is expected to increase to 578 million by 2030 [1]. There are two major types of diabetes mellitus: type 1 and type 2. Type 2 diabetes is the most common of all diagnosed cases, affecting approximately 90% of individuals [2]. It is a progressive condition, often asymptomatic and categorized mainly by insulin resistance. Microvascular and macrovascular complications are common in patients with uncontrolled diabetes.

The main cause of deaths in Type 2 diabetic patients is cardiovascular disease, with coronary heart disease and stroke being the major contributors [3]. The long-term hyperglycemic condition results in alterations in cell membrane permeability to cations and transmembrane potential [4]. Hyperpolarization, due to constant oxidative stress in diabetic cells, is responsible for the long-term complications in diabetes [5,6,7]. Sozmen et al. [8] reported uncontrolled diabetes to be responsible for glucose auto-oxidation, non-enzymatic protein glycation, and activation of the polyol pathway with increased oxidative stress. There is currently no cure for the disease. The treatment goals for diabetes mellitus include the restoration of the normal metabolic pathways and reversal of dyslipidemia caused by the disease. Conventional treatment includes a combination of oral glycemic agents, daily insulin injections, regulation of diet, or physical activity. Exercise is associated with improved insulin sensitivity and can acutely lower blood glucose in diabetic patients, as well as improve cardiovascular status. The cost associated with the treatment of diabetes is high, and it increases if there are complications—hence the urgent need to find potent and cost-effective options for the treatment of the disease. Many people use medicinal plants for the management of the disease because of the rising cost of orthodox treatment and the associated side effects. More than 800 medicinal plants around the world have been identified as possible treatment options [9]. Unfortunately, the use of most medicinal plant supplements in the management of diabetes and other diseases lack scientific backing. Inositol hexakisphosphate (IP6) and inositol supplements have been reported to regulate insulin secretion. However, current research focuses on inositol and IP6 combination in the effective management of type 2 diabetes mellitus.

## 2. Inositol

Inositol is a saturated polyol with a six-carbon ring structure where each carbon is hydroxylated. They are isomers of hexahydroxy-cyclohexanes with nine possible geometrical forms, seven of which are optically inactive, and the remaining two form a chiral pair [10]. Some are biologically active, with the most common and most stable being myo-inositol [11,12]. Myo-inositol is water-soluble and found in a variety of food products [13]. Different safe doses of inositol have been reported in the literature. A daily oral dose of 18 g of inositol for three months has been reported to be safe and well-tolerated [14]. Others have suggested that myo-inositol is safe up to doses of 12 g per day [15]. Clements and Darnell [16] observed that the greatest amounts of myo-inositol were present in fruits, beans, grains, and nuts. Myo-inositol serves as the backbone and precursor of other inositol phosphates. It is produced in the human body from d-glucose and is present in all living cells as phosphatidylinositol and phytic acid [17]. It plays important physiological roles, which include mediation of osmoregulation, anticancer activity, and the enhancement of the anticancer effects of IP6 on various cancers [18,19,20]. It is also involved in the regulation of insulin release from the pancreatic beta-cells [21,22,23,24].

## 3. Myo-Inositol Hexakisphosphate

Myo-inositol hexakisphosphate (IP6), or phytic acid, is a natural organic phosphorus compound that is present in almost all plant and mammalian cells and is the phosphorus reservoir in all grains and oilseeds [25]. It is found in food sources high in fiber content, with the most abundant sources being wheat bran and flaxseed (0.4–6.4%) [26,27]. In most cereal crops, IP6 is the primary source of phosphorus. It possibly accounts for 65–85% of the total phosphorus in seeds, with the remaining phosphorus in the form of soluble inorganic phosphate and cellular phosphorus found in macromolecules such as nucleic acids, proteins, lipids and sugars [28,29]. 

The antinutrient nature of IP6 has been described in some studies [30,31]. However, emerging research shows that IP6, as well as the lower forms of inositol phosphates, IPs 2–5, may have essential physiological functions as well as anti-inflammatory and anticancer properties [32,33,34,35]. Recent studies have ascribed antioxidant and anti-diabetic properties to this group of compounds [36,37]. IP6 is produced within cells by de novo synthesis [38]. It is biosynthesized via two different routes, the lipid-dependent and lipid-independent pathways [39,40]. In plants, the lipid-dependent pathway is prominent in all plant organs, and the lipid-independent pathway is the more prominent of the two pathways in seeds only [41]. d-glucose-6-phosphate is initially converted to myo-inositol 3-monophosphate by the enzyme myo-inositol 3-phosphate synthase. In the lipid-independent pathway, the myo-inositol 3-monophosphate undergoes a series of sequential phosphorylations through the action of various inositol phosphate kinases (Figure 1). For the lipid-dependent pathway, the myo-inositol 3-monophosphate is initially converted to myo-inositol, which is then converted to various phosphatidyl inositols, eventually leading to the formation of the higher inositol phosphates, IP5 and IP6 (Figure 1)

The involvement of phosphorylated inositols (IP1 to IP6), especially IP6, in insulin secretion has been reported [42,43]. An increase in the intracellular concentration of d-*myo*-inositol 1,2,3,4,5,6-hexakisphosphate (InsP6) in cells that are involved in insulin secretion and other cell types has been reported [44,45,46]. IP6 administration chelates iron, thereby removing it from circulation, and prevents the generation of iron-driven hydroxyl radicals as well as the suppression of lipid peroxidation [47,48]. IP6 supplementation is also believed to alter enzyme activity via direct binding. On the other hand, there are concerns about the potential adverse effects of IP6 supplementation, especially on electrolyte balance. IP6 is degraded during gastro-intestinal digestion in humans [27]. It is generally assumed that IP6 cannot cross the lipid bilayer of the membrane because of its extremely high charge density [27]. However, Sakamoto et al. showed the distribution of radioactivity in rat organs administered radiolabeled IP6 [49]. Grases et al. reported significantly higher IP6 in the plasma of rats fed 1% sodium phytate [50]. A strong effect of high IP6 intake on its concentration in plasma and brain has also been reported [51]. Sanchis et al. reported that 3-month IP6 diet consumption significantly reduced the levels of circulating advanced glycation end products in patients with type 2 diabetes mellitus alongside a 3.8% decline in HbA1C, which they presumably attributed to reduced protein glycation [52]. Da Silva et al. demonstrated the protective effect of IP6 on intestinal oxidative stress through its capacity to mitigate lipid peroxidation [53]. Studies have also shown that patients treated with an IP6 diet had a significant increase in urinary inositol phosphates (IPs) [53,54]. Grases et al. noted that when IP6 was absent from the diet, the urinary excretion of IPs declined to undetectable levels [55]. However, increasing IP6 consumption resulted in increased urinary excretion of IPs in rats. A similar trend was also noted in humans. They concluded that increased consumption of IP6 results in increased urinary excretion of total and mixtures of multiple different IPs. Although evidence supporting the absorption of IP6 exists, the mechanism of absorption is not clear. Although a specific transporter protein is yet to identified, it has been suggested that IP6 enters cells by a pinocytotic mechanism followed by subsequent dephosphorylation to InsP4 and InsP5, which are in turn, able to exert physiological effects intracellularly [56]. A divergence of views related to the absorptivity of intact IP6 is discussed elsewhere [57].

The combination of IP6 and inositol has been known to be effective in cancer treatment and is frequently reported today [58]. Shamsuddin et al. [19] demonstrated that the combination of IP6 and inositol in an appropriate ratio produces inositol 1,4,5-trisphosphate signaling molecules, which are essential cellular regulators. They concluded that a combination of IP6 and inositol might be more effective in the maintenance of human health than each component alone [58,59]. Diabetes mellitus being a disease with complex pathophysiology, effective therapy may require a multi-target, multi-effect approach to prevent complications. This review, therefore, provides an overview of our findings on IP6 and inositol supplementation in the effective management of diabetes mellitus in animal models of type 2 diabetes.

## 4. Hypoglycemic Activity of IP6 and Inositol Supplement

Shamsuddin et al. [60] reported the dephosphorylation of IP6 in cells to form lower inositol phosphates (IP1–IP5). Some researchers have postulated that IP6 may be transported into the gastrointestinal epithelial cells intact, where it is dephosphorylated [49]. The intracellular concentration of lower inositol phosphates has been shown to increase with IP6 treatment [61,62]. It is suggested that this increase may be due to the conversion of IP6 to lower inositol phosphates in the gut by enzymes as opposed to the absorption of intact IP6 [63]. Regardless of the mechanism, it appears that IP6 (probably via degradation to lower inositol forms) and inositol serve as reservoirs for the formation of lower inositol phosphates and phosphatidylinositol that could enhance their antidiabetic activities. Dilworth et al. [64] reported the lowering of blood glucose by IP6 supplementation. IP3, one of the lower inositol phosphates, has been shown to play vital regulatory roles in Ca^2+^ mobilization [32,65,66] and, therefore, insulin secretion. Lee et al. [67] reported a reduction of blood glucose concentration in diabetic KK mice administered IP6 without affecting the fasting insulin concentration. Omoruyi et al. [68] also reported that the percentage spike in random blood glucose was lowest in diabetic rats fed an IP6 supplement and attributed it to reduced intestinal amylase activity. In our study with inositol and IP6 supplement, we noted a significant reduction in the non-fasting blood glucose concentration compared with the untreated diabetic group. We also observed that serum glycated hemoglobin concentrations were not significantly altered among the diabetic groups, which may be due to the short-term duration of the study [36]. We attributed the hypoglycemic activity of the combination supplement to the significant reduction in intestinal weight, with associated alteration of the activities of some enzymes involved in carbohydrate digestion. Other studies have reported impaired starch digestion and reduction in endogenous carbohydrase activity by IP6 supplementation [36,69,70,71]. Our study also showed a significant reduction in the proximal intestinal amylase activity in type 2 diabetic rats, thus suggesting that the consumption of inositols may step-down intestinal carbohydrate digestion, which, in turn, leads to reduced blood glucose levels [36].

## 5. Inositol and IP6 Supplementation and Serum Leptin

Several systemic peptides are associated with appetite, food intake, and body weight regulation. It has been reported that plasma leptin levels decrease during fasting and increase postprandially [72]. Activation of the leptin receptor initiates various signaling cascades, which are linked to alterations in food consumption [73,74]. Ahima et al. [75] showed that a decrease in leptin action promotes an increase in the drive for food, reduced satiety, and energy utilization. Reduction in the metabolic rate has been linked to diminished liver insulin sensitivity, increased liver glucose production, and decreased glucose uptake into the muscle [76,77]. It has been demonstrated that leptin released from fat cells of the adipose tissue sends signals to the hypothalamus in the brain, which results in reduced food intake and increased energy expenditure [78,79,80,81]. Obese patients are believed to be leptin resistant; hence, their treatment focus includes overcoming leptin insensitivity. Other studies have also reported the roles of IL-6, TNFα, leptin, adiponectin, and angiotensin in childhood obesity [82,83,84,85].

On the other hand, adiponectin level is believed to increase during fasting with an associated increase in food intake. Reduced adiponectin levels promote increased production of pro-inflammatory proteins like IL-6 and C-reactive protein. These pro-inflammatory proteins have been shown to have a strong correlation with coronary artery disease [86,87]. Ghrelin, derived from the stomach, has also been shown to regulate food intake and body weight [88]. Metreleptin, a recombinant leptin analog, is indicated for treating the complications of leptin deficiency in patients with generalized lipodystrophy [89,90,91]. Nagayama et al. [92] reported that metreleptin supplementation improved serum triglyceride and lipoprotein profiles with subsequent amelioration of disturbed insulin sensitivity. Omoruyi et al. [68] noted the highest food intake in diabetic rats fed IP6 supplement that was not transformed into weight gain and concluded that the supplement might interfere with food digestion and absorption. However, in our study with inositol and IP6 supplementation in type 2 diabetic rats, we suggested that the observed significant increase in serum leptin may be a possible mechanism by which food intake and body weight gain were significantly reduced [36].

Seufert [93] reported that leptin increases insulin sensitivity, reduces hepatic glucose production, and decreases glucagon levels. Insulin is known to upregulate leptin production and secretion in adipose tissue. Soderberg et al. [94] showed that high leptin levels are associated with the future development of diabetes in Mauritian men, which is supportive of the conflicting reports on the role of leptin in diabetes in humans and animal models. Other research showed that leptin therapy improved hyperglycemia and insulin resistance in a mouse model of type 2 diabetes [95]. Leptin gene therapy has been reported to improve type 1 and type 2 diabetes and diet-induced obesity in animal models [96,97,98]. We hypothesized that inositol and IP6 supplementation might promote insulin sensitivity by increasing leptin levels in the blood [36]. The observed significant decrease in insulin resistance and reduced food intake may contribute to the hypoglycemic activity of inositol supplementation in type 2 diabetic rats. Supplementation with IP6 and lower inositols may also accrue benefits to diabetics by reducing the risks associated with obesity and other downstream pathological complications. The reduction in serum lipids, along with reduced blood glucose and enhanced insulin sensitivity in the combination treatment was consistent with the in vitro findings reported by other researchers [99,100]. Hence, treatment with inositol supplements may be effective in controlling dyslipidemia and obesity that are associated with type 2 diabetes mellitus.

## 6. Inositol and IP6 Supplementation on Complete Blood Count

Red cell distribution width (RDW) is a numerical measure of the amount of variability in the RBC size, which is routinely used in the differential diagnosis of anemia [101]. Red cell distribution width has been suggested as a predictor of cardiovascular diseases and anemia [102,103,104]. A link between high levels of RDW and the risk of developing cardiovascular disease and nephropathy has been reported [105,106]. The impairment of erythropoiesis is believed to be linked to RDW increase due to chronic inflammation and oxidative stress. Inflammation and oxidative stress contribute to the development of type 2 diabetes and associated complications [107]. Weiss and Goodnough [108] reported that inflammation might increase RDW levels through the impairment of iron metabolism thereby preventing the production of red blood cells or reducing red blood cell survival. Erythrocyte survival may be reduced by oxidative stress by promoting premature erythrocyte circulation [109,110]. Cakir et al. [111] reported significant increases in RDW values in type 2 diabetic patients. Hence, lowering blood glucose as well as reducing oxidative stress with its associated diabetic complications are essential in the effective treatment of the disease [109]. Bhowmik et al. reported that the hematological profile of mice treated with IP6 were within normal range [112]. Our study with inositol and IP6 combination showed a significant decrease in RDW value compared to untreated diabetic rats. The observed significant increase in RDW values in untreated diabetic rats may be indicative of their risk of developing vascular diseases. Our observed dyslipidemia supports this proposition in untreated type 2 diabetic rats [113]. Hence, the observed reduction in RDW levels in type 2 diabetic rats treated with inositol and IP6 supplement may accrue benefits through the protection of diabetics against the development of vascular diseases.

## 7. Inositol and IP6 Supplementation and Blood Lipids

IP6 supplementation has been shown to reduce serum triglycerides and increase total cholesterol and HDL cholesterol levels in diabetic rats [68]. Other researchers have reported the reduction in lipase activity, total cholesterol, low-density lipoprotein, hepatic total lipids, and hepatic triglycerides while increasing high-density lipoprotein levels by IP6 supplementation [114,115]. Omoruyi et al. [68] ascribed their observed increase in cholesterol to the use of calcium-magnesium phytate rather than sodium phytate used in other studies. In an in vitro study carried out by Yuangklang et al. [116], calcium phytate complex was shown not to bind bile acids. They suggested that it may reduce fecal bile excretion and increase serum cholesterol in vivo. However, reduced triglyceride and increased HDL cholesterol levels are desirable in the management of diabetes. The salt of IP6 supplement is crucial to avoid an increase in serum total cholesterol, which may be a drawback in the use of IP6 supplement in the management of diabetes. Omoruyi et al. [68] also noted a link between decreased body weight and increased HDL levels in the serum of rats fed IP6 supplement. They concluded that a decrease in body weight associated with IP6 supplementation might be sensitive to elevated serum HDL cholesterol. However, in our inositol and IP6 supplement study, we attributed the observed reduced food intake and body weight to the significant increase in the serum leptin levels [36]. We also noted that inositol and IP6 supplementation significantly decreases serum total cholesterol and triglyceride levels. We partly ascribed these changes in serum total cholesterol and triglyceride concentrations to the reported increase in intestinal lipase activity in rats treated with IP6 supplement [36,117]. We also reported an increase in fecal cholesterol and triglyceride levels by inositol and IP6 supplementation, which may further explain the observed reductions in serum cholesterol and triglyceride concentrations in type 2 diabetic rats [36]. Hence, inositols, by way of IP6 and inositol supplementation, may be effective in the management of type 2 diabetes mellitus through their ability to lower cholesterol and triglycerides, along with increased HDL-cholesterol levels.

## 8. Inositol and IP6 Administration and Organ Function

Alanine aminotransferase is a cellular enzyme that is present in low concentrations in serum under normal conditions. Significant elevation in serum concentrations of alanine aminotransferase is associated with either an increase in its synthesis or an increase in leakage from damaged cells [118]. Omoruyi et al. [68] reported a significant increase in serum alanine aminotransferase activity by IP6 supplementation. This may be indicative of liver damage, which is linked to both streptozotocin-induced diabetes and 4% level supplementation of IP6. They also reported a significant increase in the serum levels of IL-1β in the group fed IP6 supplement and suggested that the significant elevation of serum IL-1β may be due to hepatic hypertrophy with subsequent leakage of the enzyme into the blood. Damage or obstruction of the biliary tree, pancreas, liver or gall bladder promotes the release of alkaline phosphatase (ALP) into the blood. However, inositol administration via inositol and IP6 supplementation did not significantly alter serum ALP activity. Histopathological assessment of the liver sections did not show any damage to the liver by inositol and IP6 supplementation, which may also explain the observed non-significant effect of the combination on serum alanine and aspartate aminotransferases. The estimation of renal health includes the assessment of blood urea nitrogen and uric acid concentrations. Increased blood urea nitrogen is associated with renal disease or failure, congestive heart failure, dehydration, and fever. We noted that inositol supplementation was effective at modulating serum urea nitrogen concentration towards normal. Omoruyi et al. [68] also reported no adverse effect of IP6 supplementation on renal function.

Interestingly, IP6 supplementation alone or inositol and IP6 combination resulted in a downward trend in serum uric acid levels, which may be indicative of a lower risk of gout development. Previous studies have reported an increase in kidney lipid peroxidation in diabetes-induced rats [37,119]. A balance between oxidants and antioxidants in normal physiological conditions prevents tissue damage associated with free radical toxicity. Increases in the concentrations of reactive oxygen species (ROS) are associated with lipid peroxidation, protein oxidation, DNA damage, enzyme inhibition, and activation of the programmed cell death pathway [120]. Excessive generation of ROS in oxidative stress is prevented by the cellular utilization of available antioxidants to keep the levels of ROS below the stress threshold. However, IP6, as an antioxidant, is believed to curtail ROS production and confer protection against oxidative damage [121] and diseases associated with oxidative stress. We noted a significant increase in reduced glutathione (GSH) and catalase (CAT) levels by inositol and IP6 supplementation. Hence, treatment with inositol and IP6 supplement may promote renal health through an increase in antioxidants with associated normalization of lipid peroxidation. The hypoglycemic property of inositols can lead to reduced generation of oxidative products, which may ultimately lower oxidative stress. The observed reduction in oxidative stress and an increase in antioxidant activity by inositols consumption may accrue benefits to diabetics by lowering lipid peroxidation and preventing tissue damage. Overall, supplementation of the diet with inositols may be beneficial in the prevention of diabetic complications, since it reverses the diabetes-induced downregulation of the antioxidant defense system [113].

## 9. Intestinal Enzymes and Inositol and IP6 Supplementation

IP6 is believed to form a complex with proteins. It may negatively affect digestive enzymes in the gastrointestinal tract with its overall impact on nutrient absorption [122,123,124]. The enzyme amylase is synthesized and secreted by the pancreatic exocrine system and salivary glands. It is involved in the breakdown of starch and glycogen to maltose and oligosaccharides. We reported a significant decrease in serum α-amylase concentration in untreated type 2 diabetic rats [121]. Other researchers have also reported a decrease in serum α-amylase activity in diabetic patients that may be due to insulin insufficiency [125,126]. Other studies have also suggested that low serum α-amylase levels may be due to metabolic abnormalities that are attributable to a diminished pancreatic exocrine–endocrine relationship [127,128]. IP6 supplementation has been reported to significantly reduce intestinal amylase activity [68,129]. Hence, the antidiabetic functions of IP6 supplementation may be due to the reduction in intestinal amylase activity. Reduced intestinal amylase activity is associated with lesser products of carbohydrate digestion and absorption [130]. We noted an increasing trend in serum a-amylase activity in type 2 diabetic rats treated with an inositol and IP6 supplement. The increasing trend in serum α-amylase activity may account for the significant decrease in insulin resistance [36]. Adenosine triphosphatases have been implicated in the active transport of minerals across the mucosal membrane and energy production. We reported a decreasing trend in Na^+^/K^+^ ATPase activity by inositol supplementation, which may partly account for its hypoglycemic activity [122]. Our finding was similar to the report by Dilworth et al. [64] that suggested a decrease in the absorption of digested products in rats administered IP6 extracted from sweet potato. The decline in absorption might be due to a decrease in intestinal mucosa Na^+^/K^+^ ATPase activity with a subsequent reduction in blood glucose levels [53].

## 10. Inositol and IP6 Supplementation and Carbohydrate and Lipid Metabolism

Onomi et al. [131] reported reductions in liver glucose-6-phosphate dehydrogenase and malic enzyme activities by dietary phytate supplementation in rats fed a high-sucrose diet. They reported a non-correlation of liver lipogenic enzyme activity with serum concentrations of lipids and suggested that serum lipids may not be sensitive to IP6 action. Dietary phytate supplement may be protective against fatty liver via a mechanism that includes depression in lipogenesis without affecting serum lipids. However, we noted the enhancement of liver glucose-6-phosphate dehydrogenase activity in diabetic rats fed inositol and IP6 supplement, which may partly account for the hypoglycemic effect of the combination supplement. Dilworth et al. [64] also reported an increase in the activity of glucose-6-phosphate dehydrogenase activity in rats treated with IP6 extracted from sweet potato. The generation of NADPH by glucose-6-phosphate dehydrogenase is used by glutathione reductase to maintain reduced glutathione levels. We noted a decrease in the insulin resistance score (HOMA-IR) among type 2 diabetic rats treated with inositol and IP6 supplements. We proposed that the up-regulation of the glucose-6-phosphate dehydrogenase activity by inositol supplementation could be associated with the observed reduction in insulin resistance, which increases the influx of glucose into the pentose phosphate pathway and ultimately lowers the blood glucose levels.

## 11. Conclusions

The hypoglycemic activity of inositols by way of IP6 and inositol supplements may partly be due to intestinal weight reduction that results in the alteration of the activities of some enzymes involved in carbohydrate digestion. Hence, the consumption of these supplements may step down intestinal carbohydrate digestion, which in turn leads to reduced blood glucose levels. Supplementation with inositols further promotes a significant increase in serum leptin, which results in food intake reduction and body weight regulation. The decrease in insulin resistance and food intake by the supplements may also contribute to the hypoglycemic activity. The reduction in serum lipids, along with reduced blood glucose and enhanced insulin sensitivity, in the combination treatment is desirable in the effective management of the disease. Hence, the inositols produced by an IP6 and inositol combination supplement may be effective in controlling dyslipidemia and obesity that are associated with type 2 diabetes mellitus. The reduction in RDW values by inositols may accrue benefits through the protection of diabetics against the development of vascular diseases. Additionally, the increase in fecal cholesterol and triglyceride levels by inositol and IP6 supplementation may further explain the observed reductions in serum cholesterol and triglyceride concentrations in type 2 diabetic rats. Hence, inositol supplementation may be effective in the management of type 2 diabetes mellitus through its ability to lower cholesterol and triglycerides, along with the elevation in blood HDL-cholesterol levels.

The reduction in oxidative stress and an increase in antioxidant activity by the supplement may be beneficial to people with diabetes by lowering lipid peroxidation and preventing tissue damage. Supplementation with IP6 and lower inositols may be beneficial in the prevention of diabetic complications since it reverses the diabetes-induced downregulation of the antioxidant defense system. The increasing trend in serum a-amylase activity in type 2 diabetic rats treated with these supplements is indicative of the restoration of some metabolic abnormalities associated with endocrine–exocrine function in diabetes. The decrease in Na^+^/K^+^ ATPase activity by the supplement may also partly account for its hypoglycemic activity. The decline in absorption might be due to a decrease in intestinal mucosa Na^+^/K^+^ ATPase activity with a subsequent reduction in blood glucose levels. The up-regulation of the glucose-6-phosphate dehydrogenase activity by the supplement could be associated with the reduction in insulin resistance, which increases the influx of glucose into the pentose phosphate pathway and ultimately lowers the blood glucose levels. We propose that the hypoglycemic activity of inositol supplements can lead to reduced generation of oxidative products, and may ultimately lower oxidative stress. Overall, the benefits of inositols generated from IP6 and inositol combination treatment cannot exclusively be attributed to IP6 or inositol, because other IPs may also contribute to the reported beneficial effects.

## Figures and Tables

**Figure 1 molecules-25-01720-f001:**
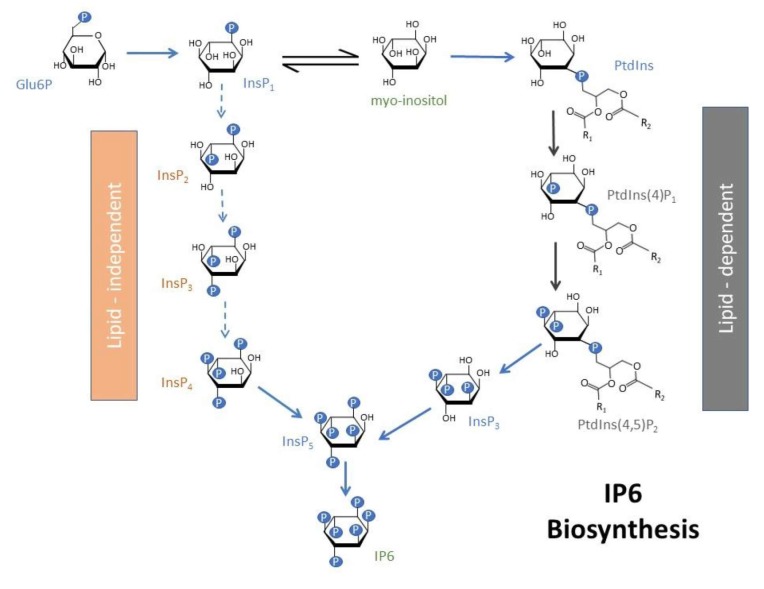
A proposed pathway for inositol metabolism involving lipid-dependent and lipid-independent pathways leading to the production of IP6.

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
