# Peer review of "New Frontiers for the Use of IP6 and Inositol Combination in Treating Diabetes Mellitus: A Review"

_molecules, 2020, doi:10.3390/molecules25071720_

Round 1
Reviewer 1 Report
This is a well written, well organized review on the physiological effects of oral inositol combined with IP6, two metabolites linked to several human diseases, including diabetes, as highlighted in the review. The major problem with this review is there is very little data supporting the underlying premise - which is synergy exists between the physiological efffects of IP6 and inositol provided in the diet. This reviewer has looked through the cited literature and cannot find data to support this. It is far more feasible that IP6 is dephosphorylated in the gut and inositol is adsorbed, a process which is well known to occur in all tested cell types and in animal intestinal epithelium.
However, since the bulk of the review is rightly focused on the physiology not mechanism, if the premise of "synergy" is removed and replaced simply with "inositol action" throughout the text, that would largely satisfy this reviewer.
Major concerns:
1. I have not seen strong evidence that inositol and IP6 synergize with each other exerting a physiological affect through distinct mechanisms, required for IP6 and inositol synergy. The authors cite very old papers (some over 30 years old that have gathered only 12 citations) as primary evidence, unless I missed something. Please do point this out if I have. Since the premise of the review is that inositol and IP6 operate through distinct mechanisms and thus synergize with each (operating through distinct mechanisms, such that just increasing the dose of one or the other cannot do the same thing as adiminstering both), this is a rather large concern. This reviewer many be ignorant of some data so if clear evidence showing distinct mechanisms of inositol and IP6 (such as gut absorption of non-hydrolyzable IP6 analogs synergizing with inositol) were specifically provided, I would be happy to revisit this criticism.
2. Even if there is synergism, it is not clear from the literature cited that intact, phosphorylated IP6 can even enter the intestinal epithelium at all. Obviously this would require a specific transporter which, to my knowledge, has not been identified. Cited Reference 49 (from 1993 with only 10 citations in that time) does not show that to be the case. The authors use quite strong language to describe this, so softening the language a great deal might be acceptable. Again, if I am ignorant of some data I would gladly reconsider this criticism. Refs 16, 46, 47, 49 do not demonstrate uptake of IP6, just uptake of tritium from labeled IP6 which is not the same thing.
3. Fig 3 is not accurate - IP3 is also classically produced from the lipid pathway by hydrolysis of PI(4,5,)P2 into IP3 for calcium signaling. Given the nature of the review, cartoons should be replaced with actual chemical structures for each intermediate.
minor concerns:
1. Line 85 has a typo
2. Lines 60-62 need references for these statements, "millions" is vague and seems quite arbitrary.
Author Response
Major concerns
Concerns 1 and 2
We are aware that the ability of Ip6 to enter cells is a debated topic. Although a specific transporter protein is yet to identified, some researchers suggest that a pinocytotic mechanism may play a role in cells’ ability to uptake IP6 (Ferry et al. 2002). This, however, does not confirm synergism and the authors are not trying to suggest this. In addition, other researchers have suggested that IP6 exerts its effects by dephosphorylation to lower analogs in the GI tract which are then able to exert their effects (Vucenik, 2019). The authors also note that the controversy surrounding the ability of IP6 to enter cells is discussed otherwise (See reference by Vucenik, 2015, and also included in the article). The authors have included opposing views on the matter in sections 3 and 4 of this article.
We note the reviewer’s concern regarding the implication of synergy between Ip6 and Inositol. Although there are varied opinions on the matter, our aim was to put forward suggestions from a physiological standpoint regarding the anti-diabetic activity of inositol and or IP6 rather than suggesting a mechanism of action related to the ability of IP6 to enter cells or the synergistic effects of these two molecules. We believe that more research is needed in this area in order to arrive at a more definitive conclusion. As suggested by the reviewer, the authors have therefore softened the language in parts (see sections 3 and 4) so as not to no give the impression of synergism as this was not our focus.
Concern 3:
See the new figure inserted as suggested by the reviewer (Line 109).
Minor concerns
- Typo was addressed
- Reference was included in section 1 as suggested (See reference by Narhe et al., below).
References
Narhe, S.; Kshirsagar, S.S.; Patil, V.S. Review on Medicinal Herbs Used for Diabetes. IJPMR. 2018, 10(8):224-228.
Ferry, S.; Matsuda, M.; Yoshida, H.; Hirata, M. Inositol hexakisphosphate blocks tumor cell growth by activating apoptotic machinery as well as by inhibiting the Akt/NFκB-mediated cell survival pathway, Carcinogenesis, 2002, 23 (12), 2031-2041. https://doi.org/10.1093/carcin/23.12.2031
Vucenik, I. Conundrum of IP6. Open Biol. 2015, 5(11). pii: 150048. doi: 10.1098/rsob.150048.
Vucenik, I. Anticancer Properties of Inositol Hexaphosphate and Inositol: An Overview. J. Nutr. Sci. Vitaminol. 2019;65(Supplement):S18-S22. doi: 10.3177/jnsv.65.S18.
Reviewer 2 Report
The paper presents a review of the use of the combination of IP6 and inositol in the treatment of diabetes. Obviously the topic is of interest, but several aspects should be clarified before publication.
- Inositol is a component that is found in an important way in the human diet. For this reason, section "2. Inositol" should include some comment on the ordinary doses consumed by humans as well as the "increases" that their supplementation implies. It is also known that humans are capable of synthesizing inositol, so it should also be noted that increases in circulating inositol involve supplemented doses.
-Section "3. Myo-Inositol hexakisphosphate" describes several studies on the generation of intracellular IP6, and on the action of IP6 on iron chelation. However, no recently published papers are cited showing that oral administration of IP6 decreases the formation of advanced glycation end-products in patients with type II diabetes. The role of intracellular IP6 is indicated but the role of extracellular inositolphosphates should also be analyzed.
-In section 6 it is indicated that the combination of IP6 and inositol showed a significant decrease in RDW values in type 2 diabetic rats. It should also be indicated if a comparison with animals treated with IP6 alone has been made. These aspects must be indicated.
-In section 8, the authors cite a study on the potential beneficial and adverse effects of diabetic rats treated with IP6 supplements (ref 56). Three groups of rats are included in this study: a group of healthy rats with a normal diet, a group of diabetic rats with a normal diet and a third group of rats to which a supplement of 4% IP6 was added. The normal rodent diet, due to its high content in whole seeds, is very rich in IP6 (1-2%). How is it possible that an increase of only 2% of IP6, in animals that already consume 1-2% of IP6, allows observing the results obtained? Could the observed adverse effects be attributed to an excessively high dose of IP6?
Author Response
Responses
- Please see the response to query 1 regarding safe doses of inositol, in lines 70-73 and associated references
- See response to query 2 outlined in section 3, lines 119-135.
- Not many studies exist specifically comparing RDW of rats fed IP6 versus inositol however we have included a study highlighting normal hematological parameters of rats fed IP6 alone. See reference 114.
- Regarding the reviewer’s concern about IP6 overload in relation to the highlighted paper, please note that the normal and diabetic groups received a normal rat diet i.e., 100% normal rat diet. In comparison, the diet fed to the diabetic IP6 treatment group was formulated as follows: 96% normal rat diet plus 4% IP6, which translates to about a 4% increase in IP6. It is difficult to tell if increasing the dose by this small percentage could qualify as an overdose but it is an important point to consider. Ideally, a study will have to be done comparing the anti-diabetic effects of normal rat chow versus rat chow with 2% IP6 added. Nonetheless, this ingredient is normally found to be safe at these concentrations. There is also room for a study comparing the effects of varying forms of commercially available IP6 since some may be easier to metabolize in the GI tract leading to variations in bioavailability.
Round 2
Reviewer 2 Report
The paper has been adequately revised, and therefore can be published in its current form.